# Diet-Regulating Microbiota and Host Immune System in Liver Disease

**DOI:** 10.3390/ijms22126326

**Published:** 2021-06-13

**Authors:** Jung A Eom, Goo Hyun Kwon, Na Yeon Kim, Eun Ju Park, Sung Min Won, Jin Ju Jeong, Ganesan Raja, Haripriya Gupta, Yoseph Asmelash Gebru, Satyapriya Sharma, Ye Rin Choi, Hyeong Seop Kim, Sang Jun Yoon, Ji Ye Hyun, Min Kyo Jeong, Hee Jin Park, Byeong Hyun Min, Mi Ran Choi, Dong Joon Kim, Ki Tae Suk

**Affiliations:** Institute for Liver and Digestive Diseases, Hallym University College of Medicine, Chuncheon 24253, Korea; eomjunga32@naver.com (J.A.E.); ninetjd@naver.com (G.H.K.); klee5484@naver.com (N.Y.K.); epark312@hallym.ac.kr (E.J.P.); lionbanana87@gmail.com (S.M.W.); jj_jeong@korea.ac.kr (J.J.J.); vraja.ganesan@gmail.com (G.R.); phr.haripriya13@gmail.com (H.G.); yagebru@gmail.com (Y.A.G.); satyapriya83@gmail.com (S.S.); dpfls3020@gmail.com (Y.R.C.); kimhs2425@gmail.com (H.S.K.); ysjtlhuman@gmail.com (S.J.Y.); jiy25n@naver.com (J.Y.H.); jmg525@naver.com (M.K.J.); heejin773@gmail.com (H.J.P.); wooju7023@naver.com (B.H.M.); choimi316@naver.com (M.R.C.); djkim@hallym.ac.kr (D.J.K.)

**Keywords:** immune response, gut-microbiota, gut-liver axis, liver disease

## Abstract

The gut microbiota has been known to modulate the immune responses in chronic liver diseases. Recent evidence suggests that effects of dietary foods on health care and human diseases are related to both the immune reaction and the microbiome. The gut-microbiome and intestinal immune system play a central role in the control of bacterial translocation-induced liver disease. Dysbiosis, small intestinal bacterial overgrowth, translocation, endotoxemia, and the direct effects of metabolites are the main events in the gut-liver axis, and immune responses act on every pathways of chronic liver disease. Microbiome-derived metabolites or bacteria themselves regulate immune cell functions such as recognition or activation of receptors, the control of gene expression by epigenetic change, activation of immune cells, and the integration of cellular metabolism. Here, we reviewed recent reports about the immunologic role of gut microbiotas in liver disease, highlighting the role of diet in chronic liver disease.

## 1. Introduction

The etiologies of chronic liver disease vary, such as infection by viruses or bacteria, alcohol or toxic substances, excessive accumulation of fat or heavy metals, and abnormal immune responses [1]. These causes can lead to viral hepatitis, nonalcoholic fatty liver disease (NAFLD), alcoholic liver disease (ALD), toxic hepatitis, and autoimmune liver disease, and can be aggravated to cirrhosis or liver cancer after a chronic course. If liver damage occurs repeatedly due to any cause, there is a high risk of developing advanced liver disease regardless of the presence or absence of symptoms, and treatment once cirrhosis occurs is necessary [2].

Recent developments in our understanding of the gut-microbiome have changed our understanding of human diseases [3]. The human gut microbiota is an ecosystem consisting of more than 2000 species that are approximately 1–2 kg in weight [4,5]. The gut microbiota is known to have a crucial role in immunology, hormones, inflammation, and metabolism [6]. However, an overall understanding of the gut microbiome and its diversity according to location, environment, sex, and age has yet to be explained.

A major unsolved factor in understanding the immune reaction of the diet is that complex diets make it difficult to demonstrate the molecular mechanism. Whole foods contain a variety of nutrients. Pairwise interactions among diets, the microbiome, and immune reactions have been widely characterized. Host-associated microorganisms alter immunity through a number of mechanisms that are discussed in detail elsewhere [7,8]. Bacteria and fungi trigger specific immune reactions that can be either inflammatory or anti-inflammatory depending on the situation. Here we focus on immunological outcomes and confirm recent findings that analyze these diet-microbiome-immune interactions and their molecular mechanisms.

## 2. Gut-Liver Axis 

The human gut microbiome consists of a complex gene pool of over 10^10^ microbes that reside in the human intestine [9]. The gut bacterial composition mainly belongs to the phyla *Firmicutes* (79.4%), *Bacteroidetes* (16.9%), *Actinobacteria* (2.5%), *Proteobacteria* (1%) and *Verrumicrobia* (0.1%), identified by using 16S rRNA sequencing methods [10]. Interestingly, the relation between the gut microbiome and human health has been recognized by many studies in the last few decades [11]. Therefore, the gut microbiome is considered a “virtual metabolic organ” that forms an axis with almost all extraintestinal organs such as the liver, heart, brain, kidney and musculoskeletal system, but the gut-liver axis has recently attracted the most attention [12].

Classically, the gut microbiome and its metabolites go into the portal circulation through the portal vein and interfere with liver homeostasis in patients with liver diseases. In this process, specific receptors recognize metabolites of intestinal microbes and trigger an inflammatory response by activating the immune system. Bacterial translocation was defined as the passage of viable bacteria through the mesenteric lymph nodes and other sites in the intestinal lumen [13,14]. After that, the concept of BT widened to include microbial products or microbial fragments such as endotoxin [lipopolysaccharide (LPS)], lipopeptides and bacterial DNA and peptidoglycan [15]. These bacterial metabolite-dependent pathways are mediated by Toll-like receptors (TLRs) or many other pattern recognition receptors. TLR signaling gives rise to tissue damage via recognition of pathogens and/or their related metabolites, which trigger cascade signaling resulting in the production of inflammatory cytokines [16].

Moreover, bacterial translocation-induced inflammation prevents the synthesis of primary bile acids in the liver through inhibition of the bile acid-synthesizing enzyme CYP7A1 [17]. Changes in bile acid homeostasis that affect bile have been identified in liver diseases, leading to gastrointestinal inflammatory disease, diabetes, and obesity. [18]. Therefore, adequate management of the gut-liver axis could be an effective preventive measure for liver disease and may inhibit the progression of disease [11].

## 3. Diet, Microbiota, and Immune Responses in Chronic Liver Disease

Dietary factors induce changes in the composition of the gut microbiome, which are usually reflected in a decrease in beneficial species and an increase in pathogenic microbiota. Liver function can be affected by altering the gut microbiota due to the close relationship between the gut and the liver. [19]. Therefore, increasing attention is being paid to probiotics, prebiotics, and synbiotics because of the involvement of the gut microbiota in the pathogenesis of metabolic diseases [20].

Probiotics are beneficial microorganisms when ingested at specified dosages [21]. Probiotic strains should live in the duodenal environment, stimulate the immune system via the production of proteins (bacteriocins) that are antagonistic to pathogens, produce short-chain fatty acids to improve the epithelial barrier, increase anti-inflammatory action, and balance the gut microbiota [22].

Prebiotics are indigestible but fermentable foods that can be fermented by bacteria to promote growth [23]. Additionally, prebiotics alter intestinal microflora for host health, including endotoxin potential and modification of intestinal barrier integrity [24]. Synbiotics are a combination of probiotics and prebiotics that are more efficient at modulating the gut microbiota than treatment with probiotics or prebiotics alone [25].

### 3.1. Nonalcoholic Liver Disease

Worldwide, for the past 20 years, NAFLD has been the most common cause of chronic liver disease [26]. NAFLD is a disease characterized by the overaccumulation of fat in hepatocytes in individuals who do not consume large amounts of alcohol [27,28]. NAFLD, which represents a variety of liver abnormalities ranging from fatty liver to nonalcoholic steatohepatitis (NASH), takes many forms and can lead to cirrhosis and liver cancer [29].

The specific pathogenesis of NAFLD is uncertain. The main pathogenesis known to date, that is, hepatic lipid overaccumulation, and insulin resistance are suggested as the main factors causing steatosis. Lipid peroxidation following oxidative stress, mitochondrial and adipokine dysfunction, and the action of proinflammatory cytokines (e.g., tumor necrosis factor [TNF]-α) also cause NAFLD or a severe form of NASH [30]. Recent evidence has explained the implications of gut-derived endotoxins and gut microbiota that actively contribute to NAFLD pathological physiology due to tight anatomical and functional crosstalk between the liver and gut. Nutrition, obesity, and environmental causes can change intestinal permeability, creating a microenvironment favorable for bacterial overgrowth, mucosal inflammation, and translocation of invasive pathogens and harmful byproducts, which in turn affects the composition of liver fat and impairs inflammatory and fibrosis processes (which affect the composition of liver fat and trigger inflammatory responses and fibrosis) [31]. Recently, NAFLD/NASH has become widespread, and it is likely to increase as the incidence of obesity and diabetes increases. However, treatment is limited to exercise, weight loss, and control of metabolic risk factors [32].

A recent review summarized the studies that show that probiotic supplementation in animal models and human studies improves inflammatory status and clinical manifestations in NAFLD [21]. When probiotics (*Bifidobacterium infantis*, *Lactobacillus acidopilus*, *Bacillus cereus*) were fed to the high-sucrose and high-fat (HSHF) diet NAFLD model, intestinal microbial and immunological changes were observed. *Lactobacillus*, *Bifidobacteria* and *Bacteroides* increased, while *Escherichia coli* and *Enterococcus* decreased. In addition, the levels of TNF-α, IL-18, serum LPS, and liver TLR4-mRNA were decreased [33]. In a high-fructose, diet-induced NAFLD model, supplementation with *L. rhamnosus GG* increased the total number of *Firmicutes* and *Bacteroidetes*. *LGG* attenuated the expression of the proinflammatory cytokines TNF-α, IL-1β and IL-8R in the liver [34]. After oral administration of *L. plantarum* for 16 weeks to a high-fat and fructose diet (HFD/F)-fed NAFLD mouse model, *Barnesiella*, *Parabacteroides*, and *Bifidobacterium* were significantly increased. In addition, *L. plantarum* inhibited the increase in serum LPS, TNF-α, IL-6, and IL-1β levels and significantly reduced the expression of the NF-κB P65 protein, a major transcriptional regulator of immune and pro-inflammatory responses, along with increased IκB protein expression [35]. In another rat model fed a high-fat diet (HFD), 6 strains of *Lactobacillus* and 3 strains of *Bifidobacterium* were injected. Combination probiotic treatment affected the gut microbiota, increasing *Bifidobacterium*, *Ruminococcus*, *Clostridium*, and *Anaerostipes*, and decreasing *Akkermansia*, *Bacteroides*, *Prevotella*, *Veillon*, *Coprococcus*, and *Roseburia*. In addition, LPS, IL-18, and IL-1β were significantly lower than those of the HFD group, and TNF-a decreased, but not significantly [36]. In another study, the HFD-induced NAFLD mouse model was supplemented with LB (*L. plantarum X, B. bifidum V*) alone or LBM (LB +Salvia miltiorrhiza Bunge polysaccharides). The results showed that *Bacteroidetes* and *Firmicutes* were increased in both groups, *Bacteroides*, *Lactobacillus*, and *Parabacteroides* were increased in the LB group, and *Cyanobacteria*, *Oscillospira*, and *Alistipes* were decreased in the LBM group. In addition, LPS, TNF-α, IL-6 and IL-1β concentrations decreased in both groups [37].

Daily administration of 5 × 10^8^ CFU *B. pseudocatenulatum* CECT 7765 to the NAFLD mouse model with HFD-induced liver steatosis increased the total number of *Bifidobacterium spp*. and decreased *Enterobacteriaceae*. It also decreased the levels of the serum inflammatory cytokines and chemokines IL-6 and MCP-1, and increased the production of the anti-inflammatory cytokine IL-4 [38]. MCP-1 plays an important role in regulating monocyte/macrophage infiltration into the liver during liver damage and sustaining liver inflammation and fibrosis [39]. *B. pseudocatenulatum* CECT 7765 restored the function of dendritic cells (DCs) damaged by an HFD, improving the ability of DCs to present antigens and stimulate T-lymphocyte proliferation [38]. Administration of synbiotics (*L. paracasei* B21060, fructo-oligosaccharides, arabinogalactan) to HFD-induced NAFLD rat model reduced gram-negative bacteria, serum ALT, AST, TNF-α, and IL-6. In addition, p50 NF-κB, TLR4, and TLR4 coreceptor CD14, which regulate the immune response, were reduced, while IκBα, which inhibits NF-κB activation, was increased [40].

In a study in which 60 NAFLD patients were randomized into probiotic and placebo groups for 12 weeks, the group receiving the probiotic mixture had significant increases in the gut microbiota *Agathobaculum, Dorea* (OTU 527923), *Dorea* (OTU 195044), *Blautia, Ruminococcus*, and *Dorea* (OTU 470168) and decreases in total cholesterol, triglyceride, and TNF-α levels [41]. In a study in which 14 patients with NASH were randomized to receive 8 g of prebiotics (oligofructose) per day for 12 weeks followed by 16 g per day for 24 weeks or placebo for 9 months, administration of oligofructose increased *Bifidobacterium* spp., *C. leptum*, and *Faecalibacterium prausnitzii*, and decreased *Clostridium cluster* XI and *Clostridium cluster* I. It also reduced LPS, IL-6 and TNF-α [42].

In this way, it can be confirmed that the intake of probiotics, prebiotics, and synbiotics in NAFLD animal models and patient studies restores gut dysbiosis and attenuates NAFLD by regulating various inflammatory factors and immune responses (Table 1). However, additional clinical studies are needed to properly identify the mechanisms by which these dietary factors affect gut microbiota and immune responses.

### 3.2. Alcoholic Liver Disease

One of the leading causes of chronic liver disease worldwide is alcoholic liver disease, after which inflammation can cause liver fibrosis that can lead to cirrhosis [43].

Long-term alcohol consumption often causes alcoholic liver disease and dysbiosis [44]. Among these, the gut microbiota plays a meaningful role in ALD and is closely related to the diseased liver through the gut-liver axis [45].

Moreover, alcohol consumption also causes enteric dysbiosis with both numerical and proportional perturbations [46]. This dysbiosis crosses the leaky intestinal barrier and affects the liver, which is the main mechanism of alcohol-related disease initiation [47]. Alcohol interferes with the intestinal-liver axis at several interconnected levels, such as the mucus barrier, including the gut microbiome epithelial barrier and at the level of antimicrobial peptide production, which causes inflammation of the liver and exposure to harmful bacteria and microorganisms [48].

Increased intestinal permeability, especially due to alcohol abuse, increases the concentration of LPS in the portal bloodstream. This induced LPS binds to TLR4 and activates NF-κB in cells, which then activates light chain potentiators, proinflammatory cytokine release, reactive oxygen species production, and oxidative stress [49].

In particular, alcohol consumption has important antifibrotic functions in the liver by suppressing natural killer cells that are cytotoxic to hepatic stellate cells (HSCs). Ethanol metabolism can interrupt cell-mediated acquired immunity by damaging proteasome function in dendritic cells and macrophages, resulting in altered allogeneic antigen presentation [50].

None of the FDA-approved drugs treat or prevent ALD. Therefore, a new treatment strategy is required for patients with ALD and a direction for improvement is needed. [51]. The results of an ALD mouse model show that all probiotic and glutamine treatments not only increased BW and occludin levels, but also dramatically elevated serum liver enzymes, TNFα, IL-6, endotoxin, and D lactate levels [45]. Likewise, in an alcohol-induced animal model, TLR4 expression was downregulated through probiotics urashiol, and Korean red ginseng, which had a positive effect on ALD treatment [52]. IL-10 is an anti-inflammatory cytokine that regulates the production of TNF-α and reduces the stimulation of LPS. In addition, IL-6 is a cytokine that prevents liver apoptosis and helps repair mitochondrial DNA in liver damage [53]. In another previous study, inulin was used to treat ALD, which is a water-soluble storage polysaccharide and is an indigestible carbohydrate called fructans [54]. LPS induces inflammatory cytokines by binding to TLR4 in liver macrophages. Dietary inulin improved ALD through inhibition of inflammatory macrophages. That is, dietary inulin attenuated alcoholic hepatitis by suppressing the mutual mechanism between LPS-TLR4- macrophages [55]. Synbiotic administration effectively reduced the plasma endotoxin and TNF-α levels and hepatic TG and increased the hepatic IL-10 level. Furthermore, synbiotic supplementation protected rats against intestinal permeability by ethanol and the number of *Bifidobacteria* and *lactobacilli* in the stool was significantly increased [56]. In this study, a substance called EGCG (epigallocatechin gallate, a major polyphenol component of green tea) was mixed with *L. plantarum* to show a synergistic effect in an animal model [57]. The formulated synbox reduced endotoxin and alcohol levels and restored liver structure. It was also shown to reduce the levels of various cellular and molecular markers, NF-kB/p50, TNF-α, IL-12/p40, and the signal transduction molecules TLR4, CD14, MD2, MyD88, and COX-2. Additionally, the microstructured synbox blocks LPS from binding to TLR4, and the TLR4/MD2 complex also blocks signaling pathways catalyzed by prostaglandin, an inflammatory mediator that causes liver damage [58]. Therefore, synbox has potential for ALD treatment and improvement.

Studies using these probiotics are being conducted in animals as well as clinical trials. This study was conducted under the conditions of an open-label, randomized, prospective clinical trial for 7 days. Also, it was conducted on 66 adult Russian males diagnosed with alcoholic psychosis and hospitalized. The control group consisted of 24 healthy non-drinking Russian males. Multiple trials of *B. bifidum* and *L. plantarum* 8PA3 in patients with alcoholic liver injury showed that probiotic supplementation was associated with a decrease in the levels of ALT, AST, GGT, LDH, and total bilirubin [59]. Several human studies over the past few years have suggested that ALD patients have proinflammatory bacteria, such as *Proteobacteria* or *E. coli* [60]. Another study conducted a randomized, controlled, multicenter study in 117 hospitalized patients with alcoholic hepatitis (57 placebo, 60 probiotics). In the probiotics group, albumin and TNF-α showed differences. In addition, the number of colonies formed by *E. coli* was significantly reduced. Therefore, the composition of microbiota flora can be changed by probiotic therapy, and probiotics may also be a potential pharmacological activity for alcoholic liver disease by inhibiting the growth of harmful bacteria [61].

A number of studies have shown that dietary elements such as probiotics and prebiotics improve alcoholic liver disease (Table 2). However, although these findings have been reported, studies on the mechanisms of intestinal microflora, liver, and diet are lacking. Accordingly, more research is needed. In addition, data on mutual safety and efficacy are essential to treat the liver using diet as a pharmacological agent.

### 3.3. Liver Cirrhosis

Liver cirrhosis is defined as the degenerate stage of regenerative nodules surrounded by fibrous bands in response to chronic liver injury, which leads to portal hypertension and end-stage liver disease [62]. Continuous hepatocyte loss and activation of hepatic stellate cells become the cause of its progression [63]. As mentioned, when fibrosis worsens, cirrhosis proceeds, which occurs for various reasons, such as NASH, higher alcohol consumption, hepatitis virus infection (hepatitis B and hepatitis C), secondary bile acids, and autoimmune diseases. Primarily, fibrosis is a natural response of the body to repair damaged hepatocytes caused by various pathophysiological processes, such as oxidative stress and the inflammatory response. These liver injuries activate quiescent hepatic stellate cells and cause fibrosis [64,65,66]. As such, HSCs play an important role in liver fibrosis. The onset of liver fibrosis results from the accumulation of an extracellular matrix, which is a highly mobile microenvironment that undergoes continuous remodeling during the recovery process [67,68]. The main cause of fibrosis is activated HSCs, but they are not the only precursors. Portal fibroblasts, fibrocytes, bone marrow cells, and liver myofibroblasts that undergo epithelial-mesenchymal transition give rise to a significant percentage of myofibroblasts in the fibrotic liver. Different cell types activate myofibroblasts depending on the etiology of liver fibrosis [69]. Intestinal microbial changes have been reported to alter our health, and cirrhosis has also been linked to intestinal microorganisms [70].

Without special treatment or improvement, this fibrosis becomes cirrhosis and hepatocellular carcinoma (HCC). Among patients with cirrhosis, two out of three patients may have the disease. To diagnose fibrosis, we mainly perform serum and ultrasound-based screening tests and fibrosis 4 scores, FibroTest/FibroSure, nonalcoholic fatty liver fibrosis scores, standard ultrasound imaging, and transient elastic contrast. Continuous direct consultation and laboratory examination ultrasound monitoring are needed to improve the disease.

Chronic damage leads to fibrosis in human organs, the progression of liver fibrosis, and serious complications such as ascites, bleeding, encephalopathy, and HCC [71]. Some strains can change the intestinal permeability, gut microbiota, and immune response; improve cognition; and help avoid serious consequences, including falls or accidents, in cirrhosis patients and animals [72,73]. Until now, before the clinical use of disaccharides and nonabsorbable antibiotics, milk and cheese and the administration of *L. acidophilus* had been utilized for the management of hepatic encephalopathy, the patents consisted only of 30 Child–Pugh B stage liver cirrhosis patients split into three randomly assigned groups [74]. In an animal model, with the combined effects of organic tax, GSH, and probiotics, the newly developed SGP and Selenium-enriched *lactobacillus* demonstrated that CCl4-induced liver fibrosis can be mitigated in mice [73,75]. Moreover, a CCl4-derived liver cirrhosis model fed *Pediococcus pentosaceus* LI05 has been shown to inhibit HSC activation, reduce the hepatic response, and degrade the expression of infection mediators and profibrogenic cytokine genes, including TNF-α, TGF-α, INOS-2, IL-6, and IL-17A [76]. Similarly, in humans, VSL#3 significantly reduced the risk of hospitalization for HE and the Child–Pugh model for end-stage liver disease scores in 15 patients with Child–Pugh B stage disease and 12 with end- stage liver disease [77,78]. Additionally, synbiotics, used to achieve a stable reduction in BZD, simultaneously reduced ammonia levels and endotoxin. Other combinations of probiotics, such as, *S. thermophilus* DSM 24731, *B. longum* DSM 24736, *B. infantis* DSM 24737, *B. breve* DSM 24732, *L. paracasei* DSM 24733, *L. acidophilus* DSM 24735, *L. delbrueckii* subsp *bulgaricus* DSM 24734, and *L. plantarum* DSM 24730 and VSL#3 improve cognitive function and reduce the risk of falls in patients with cognitive dysfunction and/or previous falls, as well as inflammation gene expression and liver function. 121 patients were evaluated for eligibility and 36 patients had cirrhosis. These randomized patients were divided into placebo groups and probiotic groups for a total of 20 weeks, with 35 patients finally analyzed [72,79]. There are also reports of improvement in endotoxin and liver function by *LGG.* A defined diet of 30 MHE and a liver cirrhosis patient with multivitamin were randomized into LGG and Placebo groups for a total of eight weeks [80,81]. In addition, probiotic treatment improves hepatic encephalopathy, one of the complications of cirrhosis, indicating that probiotics help improve cirrhosis [82,83].Probiotics and symbiotics have been closely related to cirrhosis, but the principles through which this happens and the related mechanism should be uncovered in the future (Table 3).

### 3.4. Hepatocellular Carcinoma

HCC is one of the most common malignancies in the world [84], and the second leading cause of cancer-related mortality. HCC is the most common cause of death in patients, including in liver cirrhosis [85]. The cause of the disease can be liver injury from alcohol, hepatitis virus, a high-fat diet, or cholestasis, and inflammation is an essential part of the healing response to these liver wounds. Although inflammation can promote regeneration of liver injury or induce an immune response in the short term, chronic inflammation and wound healing reactions have a great association with fibrosis, cirrhosis, and HCC. Nonparenchymal cells generally promote inflammation, fibrosis, and HCC, whereas suppression of NF-κB activation in parenchymal cells promotes HCC [86]. In patients with liver fibrosis or cirrhosis, 80% develop HCC [87]. Liver transplantation for HCC is the best treatment choice in patients with early-stage tumors and accounts for one-third of all liver transplantations performed at transplantation centers. The Milan criteria are the most common criteria to select patients with HCC for transplantation, but they can be seen as too restrictive [88]. Recently, the measurement of intestinal microbial changes has been used as a biomarker to obtain hints about whether liver cancer is present [89]. Unlike developing countries, NAFLD/NASH is increasingly the cause of HCC in developed countries or Western countries, and is related to metabolic syndrome and obesity. Intestinal microbial communities are related to the development of NAFLD/NASH [84,90].

The gut microbiota contributes to the progression of HCC through the gut-liver axis [91]. Thus, the use of prebiotics, probiotics, and synbiotics may offer new ways to treat or prevent the development of HCC through the control of gut microbiota [92].

When the HCC mouse model was treated with a probiotic mixture, the abundance of *Bacteroidetes* increased compared to the control, while *Firmicutes* and *Proteobacteria* decreased. IL-17 produced by Th17 immune cells decreased, and the anti-inflammatory cytokines IL-27, IL-13, and IL-10 increased [93]. As a carcinogen, diethylnitrosamine (DEN) is effective in inducing liver tumor formation in rodents [94]. Administration of VSL#3 probiotics in a rat model of DEN-induced liver cancer decreased the proportions of *E. coli*, *Atopobium* cluster, *B. fragilis* group, and *Prevotella*; increased LPS and IL-6 levels; and increased IL-10 levels. Therefore, it is suggested that administration of VSL#3 inhibits the progression of HCC by restoring intestinal homeostasis and reducing protumorigenic inflammation [95]. Inulin-type fructans are prebiotic nutrients that regulate host immunity and metabolism and change the composition and activity of gut microbiota. In a study in which BaF3 cells were transplanted into mice to induce malignant tumors in the liver and inulin-type fructans were administered, the intestinal microbial composition of BaF3 mice was similar to that of control mice, and there was no difference between total bacteria and gram-negative *Bacteroides*. In addition, administration of inulin-type fructans decreased IL-4, IL-8, IFN-γ, and MCP-1 levels [96] (Table 4).

There are currently few clinical trials related to HCC and probiotic intestinal drugs, and further studies should be conducted to clarify the possibility of use in HCC as a therapeutic agent.

### 3.5. Diet, the Immune System and Liver Disease

Recent research aims to improve liver disease by regulating immune responses through diet (Table 5). When an NAFLD mouse model fed a high-fat Western (HFW) diet was supplemented with green tea polyphenol epigallocatechin-3-gallate (EGCG), serum ALT and AST decreased, and TNF-α levels decreased. In addition, by reducing the genes involved in the differentiation and formation of IgA^+^ B cells, TLR4, and B-cell activating factor (BAFF), which were increased in the HFW group, the ileal immune response caused by the HFW diet was alleviated [97]. Potatoes are a food crop and are a source of antioxidants, minerals, and proteins [98]. HFD-induced rats were treated with alcalase treatment –derived potato protein hydrolysate (APPH). The results showed that hepatic fat accumulation and hepatic apoptosis were suppressed. In addition, treatment improved the survival of hepatocytes by increasing the levels of PI3K and AKT, which promote immune cell activation by regulating inflammatory cytokines [99]. In another study, 52 obese children with NAFLD were supplemented with tomato juice along with calorie-restricted regimen (RCR) for 60 days in a randomized crossover clinical trial. As a result, malondialdehyde (MDA), a marker of oxidative stress, was reduced. In addition, GSH, which plays an important role in maintaining numerous immune functions, including lymphocyte proliferation and NK cell activity, increased compared to the RCR alone group and was able to activate T cells. Therefore, lycopene-rich tomato supplementation can slow the progression of NAFLD in obese children by improving immune function and antioxidant capacity [100].

Fish oils that have long-chain n-3 PUFAs (LC n-3 PUFAs), vegetable oils containing n-3 PUFAs, and α-linolenic acid (ALA) have been used to improve liver disease with efficacy [101]. Ethanol induces CYP2E1, 3-ketoacyl-CoA, NOS, and H2O2 through oxidative stress, which is highly associated with liver disease and cancer [102]. In this study, the PUFA diet in an alcoholic animal model blocked oxidative modifications of enzymes, thereby protecting mitochondrial enzymes and preventing disorders in alcohol-induced liver disease [103]. In other studies, DHA inhibits SCD-1 and increases HO-1, of which SCD-1 is the restriction enzyme for the biosynthesis of MUFAs. In the case of deficiency, it improves fatty liver [104,105]. It also increases Ho-1, which induces antioxidant stress and is involved in cell survival [106]. Therefore, docosahexaenoic acid (DHA) helps in acute alcoholic liver disease by inhibiting SCD-1 and inflammatory cytokines [104]. Previous studies have shown that TLR4 expressed in Kupffer cells recognizes CD14 bound to LPS and triggers endotoxin-induced signaling mechanisms. Accordingly, the NF-kB signal is activated through the MyD88 pathway, and Kupffer cells induce liver damage by generating proinflammatory mediators such as cytokines and free radicals. On the other hand, flaxseed oil rich in α-linolenic acid prevents liver damage by reducing this mechanism [107].

Branched-chain amino acids (BCAAs) are characterized by altered energy metabolism and decreased protein levels in chronic liver disease. In an animal model, it can be seen that administration of BCAAs helps liver regeneration and thus plays a major role as a nutrient for improving liver function [108]. Treatment is also important in patients with severe liver disease, but nutritional care is a priority, and the use of oral supplements such as BCAA has been the subject of debate. Accordingly, BCAA oral supplementation was consistently fed to five cirrhosis patients for three weeks and a randomized study was divided into control groups and BCAA groups for 124 HCCs. When prescribed to patients with chronic liver disease, cirrhosis, or HCC, BCAAs improve the activity of neutrophils and natural killer cells in the immune system, increase albumin levels in serum, and consequently lower morbidity [109,110].

Garlic contains several compounds that can affect immunity [111]. In a randomized double-blind trial, when 42 patients with liver cancer were supplemented with aged garlic extract (AGE) for 6 months, the number and activity of NK cells in peripheral blood increased after 3 months. In addition, serum total cholesterol and HDL cholesterol levels also increased [112]. The reciprocal system between the immune system and nutrition is complex. Thus, the goal is to determine the effect of diet on the immune system, but these studies are still in their infancy, and further human and animal studies are needed.

## 4. Perspective

Since there is not yet an accurate treatment for liver disease, early diagnosis and periodic examination are important. To date, there have been many studies on improving liver disease through the regulation of the gut microbiome by supplementing the diet and intestinal drug. However, studies on which mechanisms influence immunity and gut microbiome regulation are lacking.

Clinical studies and animal model studies on liver disease are improving the treatment of liver disease by changing gut microbiota through probiotics and prebiotics to improve liver function.

However, the gut microbiota composition is different for each individual, and it is a future goal to elucidate the mechanism of which probiotic strain has the greatest effect on liver disease, and additional research is needed on individual dose administration, strain type, and number of days of administration.

Diet control, including the aforementioned probiotics, also showed an effect on the improvement of liver disease. Similarly, research should be conducted to develop a substance with pharmacological activity in the future. Few studies have found a synergistic effect using both probiotics and diet. Therefore, if we accurately understand the interaction and mechanism of these two substances, we will be able to discover more effective substances than using them alone.

Probiotics and diets are attractive frontrunners for the personalized medicine of the future. The composition and number of the gut microbiome are different because each individual’s DNA and living environment is different. Therefore, after identifying each gut microbiome, it will be possible to provide personalized liver disease treatment that involves controlling the gut microbiome by prescribing a specific diet and intestinal drug.

## 5. Conclusions

Liver disease is a serious disease caused by obesity, alcohol consumption, and viral infections. Moreover, the liver is an organ in close contact with the intestine and is affected by changes in the gut microbiota. Changes in gut microbiota through probiotics and food affect liver disease improvement. Thus, the environment of the microbiome can be changed by food and adapted and modulated by various diets and immunity, which determines human health and disease progression. However, although probiotics and diet can improve liver disease, results vary widely and show differences according to individual liver diseases such as NAFLD, ALD, Cirrhosis, and HCC. This is an innovative way to present customized medical care according to the disease, and the difference in the composition of gut microbiota will also serve as a biomarker that can determine the disease state. In addition, probiotics and diets that are specific to each disease are suggested, helping to improve liver disease and possibly to create a synergistic effect by introducing a treatment that combines prebiotics and diet.

In the future, it will be possible to diagnose and prevent diseases through AI that has built up a database. In addition, it is predicted that it will be able to help not only liver disease but also individual well-being by being presented with a personalized diet. 

There are currently few studies on these various and complex mechanisms, but many studies are being introduced, and studies that will help us better understand the microbiome will be performed in the future. Therefore, it is believed that these will lead to future approaches for using food and personalized medicine.

## Figures and Tables

**Table 1 ijms-22-06326-t001:** Animal and human studies using diet in NAFLD.

Conditions		Treatment	Main Results	Ref
Animal	HSHF diet	*B. infantis*, *L. acidopilus*, *B. cereus*	(↓) *E. coli, Enterococcus*, TNF-α, IL-18, Serum LPS, TLR4 (↑) *Lactobacillus, Bifidobacteria, Bacteroides*	[33]
	High-fructose diet	*LGG*	(↓): TNF-α, IL-1β, IL-8R (↑): *Firmicutes, Bacteroidetes*	[34]
	HFD/F diet	*L. plantarum*	(↓): Serum LPS, TNF-α, IL-6, IL-1β, NF-κB P65 (↑): *Barnesiella, Parabacteroides, Bifidobacterium*, IκB	[35]
	HFD diet	*Lactobacillus, Bifidobacterium*	(↓): *Akkermansia, Bacteroides, Prevotella*, *Veillonella, Coprococcus, Roseburia*, LPS, IL-18, IL-1β (↑): *Bifidobacterium, Ruminococcus*, *Clostridium, Anaerostipes*	[36]
		*L. plantarum* X, *B. bifidum* V Polysaccharide (Salvia miltiorrhiza)	(↓): *Cyanobacteria, Oscillospira, Alistipes, LPS*, TNF-α, IL-6, IL-1β (↑)*:* *Bacteroidetes, Firmicutes, Lactobacillus, Parabacteroides*	[37]
		*B. pseudocatenulatum* CECT 7765	(↓): *Enterobacteriaceae*, IL-6, MCP-1, IL-10 (↑): *Bifidobacterium* spp., IL-4, Induction of T-lymphocyte proliferation	[38]
		*L. paracasei* B21060, Fructo-oligosaccharides, Arabinogalactan	(↓): Gram-negative bacteria, ALT, AST, TNF-a, IL-6, p50 NF-κB, TLR4, CD14 (↑): IκBα	[40]
Human	NASH patients	Probiotics	(↓): Total cholesterol, TG, TNF-α (↑): *Agathobaculum, Dorea* (OTU 527923), *Dorea* (OTU 195044), *Blautia, Ruminococcus, Dorea* (OTU 470168)	[41]
		Oligofructose	(↓): *Clostridium* cluster XI, *Clostridium* cluster I, LPS, IL-6, TNF-α (↑): *Bifidobacterium* spp., *C. leptum, F. prausnitzii*	[42]

↑ indicates an increase in condition, ↓ indicates a decrease in condition. ALT, alanine aminotransferase; AST, aspartate aminotransferase; TG, triglycerides; TNF- α, tumor necrosis factor alpha; TLR, toll-like receptor; IL, interleukin; LPS, lipopolysaccharide; NF-kB, nuclear factor kappa-light-chain-enhancer of activated B cells; CD, cluster of differentiation; IκB, nuclear factor of kappa light polypeptide gene enhancer in B-cells inhibitor; MCP-1, monocyte chemoattractant protein 1.

**Table 2 ijms-22-06326-t002:** Animal and human studies using diet in ALD.

Conditions		Treatment	Main Results	Ref
Animal	EtOH-induced liver injury	Glutamine, Golden Bifido (probiotic mixture containing live *L. bulgaricus, Bifidobacterium*, *Streptococcus thermophilus*), Medilac S^®^ (probiotic mixture containing live *B. subtilis* and *E. faecium*)	(↓): *Proteobacteria*, *Actinobacteria* AST, ALT, TG, TNF-α, IL-6 (↑): *Firmicutes*	[45]
	High fat diet+ EtOH + LPS	*L. rhamnosus* R0011, *L. acidophilus* R0052, KRG (Korea red ginseng), urushiol (Rhus verniciflua Stokes)	(↓): AST, ALT, TNF-α, IL-1β, TLR4 (↑): IL-10	[52]
	EtOH-containing modified Lieber-DeCarli liquid diets	Inulin	(↓): *Parasutterella*, Plasma LPS Levels, TNF-α, IL-6, IL-17A, IL-10, Macrophages, TLR4 (↑): *Allobaculum**, Lactobacillus, Lactococcus*	[55]
	Ethanol by oral gavage	Synbox (*L. plantarum* + EGCG a phenolic compound)	(↓): NF-kB, TNF-α, IL-12 / p 40, TLR4 MD2, CD14, MyD88, COX-2(↑): *L. acidophilus*, SOD	[58]
	Ethanol liquid diet	Symbiotic (*L. acidophilus*, *L. bulgaricus, B. bifidum, B. longum*, Inulin, Vitamin B_1,_ Niacinamide.)	(↓): TNF-α, IL-1β, IL-6 (↑): The numbers of *lactobacilli* and *bifidobacterial*, IL-10	[56]
Human	Adult alcoholics	*B. bifidum*, *L. plantarum 8PA3*	(↓): AST, ALT, GGT, LDH, Bilirubin (↑): Restoration of bowel flora	[59]
	Alcoholic hepatitis (AH)	*L. subtilis*, *S. faecium* or placebo	(↓): *E. coli*, AST, ALT, γ-GT, ALP, LPS, TNF-α, IL-1β	[61]

↑ indicates an increase in condition, ↓ indicates a decrease in condition. ALT, alanine aminotransferase; AST, aspartate aminotransferase; TG, triglycerides; TNF- α, tumor necrosis factor alpha; TLR, toll-like receptor; IL, interleukin; LPS, lipopolysaccharide; Mψs, macrophage; ALP, alkaline phosphatase; NF-kB, nuclear factor kappa-light-chain-enhancer of activated B cells; COX, cyclooxygenase; MyD88, myeloid differentiation primary response 88; CD, cluster of differentiation; GGT, gamma glutamyl peptidase; LDH, lactate dehydrogenase; γGT, gamma-glutamyl transferase.

**Table 3 ijms-22-06326-t003:** Animal and human studies using diet in Cirrhosis.

Condition		Treatment	Main Results	Ref
Animal	CCl4	Selenium-enriched *lactobacillus*	(↓): ALT, AST, Ca2+, Phagocytic index, TNF-α (↑): GSH-Px, SOD	[75]
		*L. acidophilus*, *S. cerevisiae*	(↓): ALT, AST, α-SMA, TGF-β1, TIMP-1, Collagen-TNF-α, IL-6, MCP-1, Bcl-2, Bcl-Xl, GSH-Px (↑): Bax, SOD, GSH	[73]
		*P. pentosaceus* LI05	(↓): ALT, AST, Endotoxin, Collagen α1, TIMP-1, iNOS-2, TNF-α, IL-6, IL-17A (↑): ZO-1	[76]
Human	HCV-related Child B liver cirrhosis	*L.acidophilus* *L.helveticus* *Bifidobacteria*	(↓): Endotoxin, Ammonia level	[74]
	Cirrhosis and HVPG >10 mmHg	VSL#3	(↓): Endotoxin, Plasma renin, Aldosterone (↑): TNF-α, IL-6, IL-8	[77]
	Cirrhotic patients with MHE	*LGG*	(↓): Ammoniagenic amino acids, TNF-α, IL-13, (↑): Lachnospiraceae, SIP	[81]
	Cirrhosis with PHES less than-4	Mixture of eight strains (DSM 24731, DSM 24732, DSM 24736, DSM 24737, DSM 24733, DSM 24735, DSM 24734, DSM 24730)	(↓): Walking problems, AST, ALT, GGT, Serum bilirubin, CRP, TNF-α, FABP-6 (↑): Mean arterial pressure (mm Hg), MFI	[72]
	Cirrhosis with HE	VSL#3	(↓): CTP score, MELD score, Ammonia, Renin, Aldosterone, TNF-a, IL-1β, IL-6, Indole (↑): SBP	[79]

↑ indicates an increase in condition, ↓ indicates a decrease in condition. ALT, alanine aminotransferase; BA, bile acids; HE, hepatic encephalopathy; TG, triglycerides; VSL#3: four strains of *Lactobacillus* (*L. casei*, *L. plantarum, L. acidophilus*, and *L. delbrueckii* subsp. *bulgaricus*), three strains of *Bifidobacterium* (*B. longum*, *B. breve*, and *B. infantis*), and one strain of *Streptococcus salivarius*; GSH-Px, Glutathione peroxidase 1; CRP, C-reactive protein; SOD, superoxide dismutase; MFI, mean fluorescence intensity; SBP, spontaneous bacterial peritonitis.

**Table 4 ijms-22-06326-t004:** Animal and human studies using diet in HCC.

Conditions		Treatment	Main Results	Ref
Animal	Injection of mouse hepatoma cell line Hepa1-6	Prohep (LGG, *E. coli* Nissle 1917, heat inactivated VSL#3 (*S. thermophilus*, *B. breve*, *B. longum*, *B. infantis*, *L. acidophilus*, *L. plantarum*, *L. paracasei, L. delbrueckii* subsp.))	(↓): *Firmicutes, Proteobacteria*, Recruitment of Th17, IL-17 (↑): *Bacteroidetes*, IL-27, IL-13, IL-10	[93]
	Injection of DEN	Probiotic VSL#3 (four *Lactobacilli*, three *Bifidobacteria*, and one *Streptococcus thermophilus* subsp *Salivarius*)	(↓): *E. coli*, *Atopobium* cluster, *B. fragilis* group, *Prevotella*, *ALT*, Plasma LPS, IL-6, Nuclear NF-κB translocation (↑): IL-10	[95]
	Injection of Bcr-Abl-transfected BaF3 cells	Prebiotics (ITF)	(↓): IL-4, IL-6, IL-8, IFN-*γ*, MCP-1	[96]

↑ indicates an increase in condition, ↓ indicates a decrease in condition. ALT, alanine aminotransferase; IL, interleukin; LPS, lipopolysaccharide; NF-kB, nuclear factor kappa-light-chain-enhancer of activated B cells; Th17, T helper 17 cells; MCP-1, monocyte chemoattractant protein 1.

**Table 5 ijms-22-06326-t005:** Animal and human studies using diet in liver diseases.

Conditions		Treatment	Main Results	Ref
Animal	High- fat/Western diet	EGCG	(↓): ALT, AST, TNF-α, Genes involved in the differentiation and formation of IgA+ B cells, TLR4, BAFF	[97]
	HFD diet	APPH	(↓): Number of apoptosis nuclei, Fat accumulation (↑): p-PI3K, p-Akt	[99]
	Ethanol feeding for 9 weeks	polyunsaturated fatty acids (PUFA)	(↓): TG, Cholesterol, CYP2E1, NOS, H2O2, 3-ketoacyl-CoA thiolase	[103]
	Ethanol every 12 h for 3 administrations	DHA	(↓): TG, SCD-1, IL-6, TNF-α (↑): HO-1	[104]
	Binge liquid ethanol feeding	Flaxseed oil	(↓): AST, ALP, TG, MYD88, LPS, CD14, NF-κB p65 (↑): SOD, ZO-1	[107]
	Protein-free, calorie-rich diet	10% dextrose + 3% amino acids (35% BCAA)	(↓): Phenylalanine, Tyrosine (↑): 3H-thymidine	[108]
Human	NAFLD patients	lycopene-rich tomato juice	(↓): ALT, AST, TC, LDL, TG, IL-4, MDA (↑): HDL, GSH, T cell activation	[100]
	Cirrhosis patients	BCAA	(↑): Improve the phagocytic function of neutrophils, Natural killer activity of lymphocytes	[109]
	HCC patients Child B and C liver disease	BCAA	(↓): Morbidity rate (↑): Red Blood cell and Serum albumin levels Body weight and performance status	[110]
	HCC patients	AGE	(↑): Total cholesterol, HDL, Number and activity of NK cells,	[112]

↑ indicates an increase in condition, ↓ indicates a decrease in condition. ALT, alanine aminotransferase; AST, aspartate aminotransferase; ALP, alkaline phosphatase; TC, total cholesterol; TG, triglycerides; TNF- α, tumor necrosis factor alpha; TLR, toll-like receptor; BAFF, B-cell activating factor; IL, interleukin; LPS, lipopolysaccharide; NF-kB, nuclear factor kappa-light-chain-enhancer of activated B cells; CD, cluster of differentiation; PUFA, polyunsaturated fatty acids; DHA, docosahexaenoic acid; NOS, nitric oxide synthase; CYP2E1, ethanol-inducible cytochrome P450 2E1; SCD-1, stearoyl-CoA desaturase-1; Ho-1, heme oxygenase-1; MyD88, myeloid differentiation primary response 88; ZO-1, zonula occludens; p-Akt, phosphor protein kinase B; p-PI3K, phosphor-P13 kinase p85; MDA, malondialdehyde; GSH, glutathione; NK, natural killer; HDL, high density lipoprotein.

## Data Availability

Data are contained within the article.

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
