# Peer review of "Diet-Regulating Microbiota and Host Immune System in Liver Disease"

_ijms, 2021, doi:10.3390/ijms22126326_

Round 1

Reviewer 1 Report

In general, I found this review interesting and well written. Th authors have clearly summarized recent evidence on the relationship between diet-regulating microbiota and host immune system, especially in liver diseases. For this reason, I would suggest to clarify in the title that this review is focused on liver diseases. Moreover, I would suggest to expand the discussion section (i.e., perspective and conclusion), explaining the needs and approaches for future research.

Reviewer 2 Report

Eom et al. review the role of microbiota in liver disease. The authors provide a good summary how pro-/pre-/syn-biotic interventions can affect different types and stages of liver disease. The text is well grafted and easy to follow. Before publication, I would suggest that the authors address these comments:

  • As the focus is on liver disease, I would suggest adjusting to title accordingly.
  • For the listed human trials, please, also include the size of the study population / study groups and the trial design (e.g., RCT, crossover, …).
  • I would suggest expanding the conclusion section. The highlighted studies clearly demonstrate that treatments can be effective, but the attempted treatment regimens are very diverse. Is there anything that we can conclude on the relative effect sizes of the different treatment options and on the most promising directions for the future?
